# Fast holographic scattering compensation for deep tissue biological imaging

Molly A. May [1✉], Nicolas Barré[1], Kai K. Kummer [2], Michaela Kress[2], Monika Ritsch-Marte [1] & Alexander Jesacher[1]

Scattering in biological tissues is a major barrier for in vivo optical imaging of all but the most superficial structures. Progress toward overcoming the distortions caused by scattering in turbid media has been made by shaping the excitation wavefront to redirect power into a single point in the imaging plane. However, fast, non-invasive determination of the required wavefront compensation remains challenging. Here, we introduce a quickly converging algorithm for non-invasive scattering compensation, termed DASH, in which holographic phase stepping interferometry enables new phase information to be updated after each measurement. This leads to rapid improvement of the wavefront correction, forming a focus after just one measurement iteration and achieving an order of magnitude higher signal enhancement at this stage than the previous state-of-the-art. Using DASH, we demonstrate two-photon fluorescence imaging of microglia cells in highly turbid mouse hippocampal tissue down to a depth of 530 $\mu$m.

[1] Institute of Biomedical Physics, Medical University of Innsbruck, Innsbruck, Austria. [2] Institute of Physiology, Medical University of Innsbruck, Innsbruck, Austria. ✉email: molly.may@i-med.ac.at

Microscopic imaging inside living tissue is a key technology for understanding the functioning of life at the cellular level and its organization to form entire organs. A serious obstacle for obtaining clear vision is the tissue itself, which is highly scattering due to its often inhomogeneous structure, and thus ultimately limits the achievable imaging depth. Besides endoscopic approaches, two major strategies can be identified which have proven successful in enhancing image fidelity: i) the use of longer wavelengths to decrease scattering and ii) the use of adaptive optics to counteract aberrations.

Regarding the first point, the introduction of nonlinear microscopies such as two- and three-photon excitation fluorescence[1–4] and harmonic generation imaging[5–10] represent major improvements. These techniques rely on infrared excitation wavelengths that are less prone to scattering and provide optical sectioning by nonlinear signal generation, rendering it unnecessary to image the generated, visible wavelength signal back through the tissue onto a pinhole as in confocal microscopy. Further increased penetration depth can be achieved using high power pulses from regenerative amplifiers[11]. Progress regarding the second point has been achieved by the introduction of dynamic wavefront shapers such as deformable mirrors and liquid crystal spatial light modulators (LC-SLM) in conjunction with the development of strategies for quickly finding optimal correction patterns[12–17].

However, as imaging depths increase, so does the complexity of the wavefront scrambling, and full wavefront compensation can easily overburden the possibilities of any correction technique. The demonstration that correcting even a small fraction of the aberration can yield a single, strongly amplified speckle which acts as an imaging beam[18,19] has sparked interest in the community and led to several developments that tackle the regime of diffuse scattering.

Broadly speaking, approaches to scattering correction can be divided into direct and indirect wavefront sensing methods. The former capture the phase topography in a single snapshot, for instance by using a wavefront sensor or digital holography in combination with digital optical phase conjugation[20–24]. Such methods usually require a "guide star" or other means to tag the light originating from the desired imaging point. Guide stars can be small and strong sources such as fluorescent beads[22,25,26] or, for moderately scattering samples, simply the two-photon fluorescence generated in the focal spot itself[27,28]. Alternatively, tagging can be provided by focused sound waves[29,30] or coherence gating[31,32], similar to optical coherence microscopy. The advantage of direct wavefront sensing methods is their speed potential, which must ultimately outpace the persistence times imposed by living tissue. These vary significantly, from minutes[33] down to merely a few milliseconds[34], although a recent study highlighted the possibility to selectively correct modes of longer persistency even in quickly decorrelating biological samples[35]. In any case, speed is traded against increased experimental complexity in direct wavefront sensing systems.

Conversely, methods based on indirect wavefront sensing are often technologically much less complex, which is a very important trait regarding their use in biomedical imaging. These methods aim at finding correction patterns from a series of test measurements. Despite usually being slower, recent developments have enabled impressively fast corrections on the order of a millisecond and below using acousto-optic light modulation[36,37].

Indirect sensing techniques have also achieved notable in vivo imaging results in various tissues including the mouse brain[33,38,39]. The two techniques which have been used to achieve these results have thus attracted particular attention in the field. The first is known as iterative multi-photon adaptive compensation technique (IMPACT)[33] and is a multi-photon variant of a much earlier introduced multidither coherent optical adaptive technique (COAT)[40]. The second was introduced in 2017 as focus scanning holographic aberration probing (F-SHARP)[38] and can be viewed as an interferometric method.

Since these breakthroughs, progress in developing alternative algorithms has mostly focused on genetic algorithms[41–43] or machine learning[44,45], but their application to biological imaging has not yet been demonstrated. In addition, a direct comparison between IMPACT and F-SHARP is not available to date, which makes it difficult to judge their respective strengths and weaknesses.

The benefit of continuously updating the correction pattern during measurement has been highlighted as well[14,46]. However, existing techniques operate only on the phase level, meaning that a newly tested phase mode is directly added to the previous pattern at its optimal phase shift. Such strategies have been demonstrated for binary mode bases[46] such as Hadamard patterns[14] or patterns that are derived during the measurement process from genetic or machine learning algorithms[41–45].

Here, we introduce dynamic adaptive scattering compensation holography (DASH), a new algorithm that enables significantly faster convergence than IMPACT and F-SHARP. The primary innovation is that in DASH the correction pattern is updated in a novel way, immediately after the phase and amplitude of a mode have been interferometrically measured. This causes the signal to rise continuously, resulting in a substantial speed advantage compared to F-SHARP and IMPACT, which both pursue a stepwise implementation of the phase correction. Numerical simulations and experimental evaluations of the DASH algorithm in two-photon excited fluorescence microscopy (TPEF) reveal signal enhancement that is roughly 10x higher at the end of the first iteration compared to F-SHARP and IMPACT, although the exact enhancement ratio can vary based on experimental factors like the structure of the scatterer and the accuracy of the F-SHARP interferometer alignment.

Furthermore, the achievable imaging depth and convergence time in biological tissues are often limited by weak two-photon signals. We show numerically that DASH efficiently uses the information from each measured photon, allowing correction with about half as many collected photons and requiring half as many measurements compared to stepwise algorithms. The power of DASH under low signal to noise conditions was also demonstrated experimentally by TPEF imaging of microglia in the mouse hippocampus at depths down to 530 μm, where F-SHARP did not converge.

DASH is a continuously updating algorithm. In contrast to previous techniques, however, it employs a different update scheme, where the optimal phase found for a particular test mode is holographically combined with the previous phase pattern, i.e., their respective complex fields are added and the phase of the complex sum is taken as the new correction phase to be displayed on the SLM. As we show here, this update scheme leads to considerably faster and more robust convergence, especially in high noise environments. Our method also uses a pre-defined, non-binary test basis, which eliminates computationally intensive calculations during the measurement.

Furthermore, we develop a theoretical framework showing that IMPACT and F-SHARP are fundamentally equivalent. A notable practical difference, however, is that F-SHARP decouples the measurement speed from the SLM update rate, which enables the use of high pixel-count and cost-effective, but relatively slow liquid crystal SLMs. On the other hand, the advantage of IMPACT is the experimental simplicity afforded by its intrinsic common-path design. The equivalence of IMPACT and F-SHARP and the superior performance of DASH are further validated using numerical simulations.

A notable advantage of DASH is its generalizability to a broad range of mode bases, which could enable tailored mode selection based on noise level, fluorescence properties, and structure of the scattering object in a given experiment.

In summary, this work introduces a new scattering compensation algorithm which converges faster and with less measured photons than previous approaches using holographic interferometry to continuously improve the signal.

## Results and discussion

**Comparison to existing algorithms.** DASH was benchmarked against five other algorithms using numerical simulations for three different integration times corresponding to decreasing signal levels as shown in Fig. 1b–d. The comparison was made to F-SHARP, IMPACT, and three established continuously updating algorithms including a genetic algorithm (GA) following the procedure in ref. [41], and the partitioning algorithm (PA) and continuous sequential algorithm (CSA) described in ref. [46]. Details about the implementation of the GA are provided in Supplementary Section 5.

All simulations assume $N_{scat} = 1024$ scattering and $N = 256$ correctable pixels. Each method is tested on the same set of 10 different white noise random scatterers located in the Fourier

plane. The standard error of the mean enhancement over the 10 trials is represented by the color bands around the data. The sample is a two-dimensional homogeneous fluorescent layer in the focal plane of the objective lens. An ideal detector is assumed, with no readout noise and a quantum efficiency of 100%.

While all of the algorithms achieve some signal enhancement at the highest initial signal level of $I_0 = 1000$ photons/measurement as shown in Fig. 1b, the signal rises much more quickly during DASH than for the other algorithms and converges to a significantly higher enhancement than the other continuously updating algorithms. F-SHARP and IMPACT converge more slowly, but ultimately to the same enhancement level as DASH. Notably, they exhibit significantly higher variance in their enhancements before convergence as evidenced by the by the broad error bands on these data sets, indicating a higher sensitivity to initial conditions like the structure of the scattering mask.

For lower initial signal levels of $I_0 = 100$ photons/measurement as shown in Fig. 1c, the continuously updating GA, PA, and CSA algorithms do not converge while DASH, F-SHARP, and IMPACT converge after ~10 iterations. The poor performance of the continuously updating algorithms for low photon numbers significantly decreases their utility for biological experiments

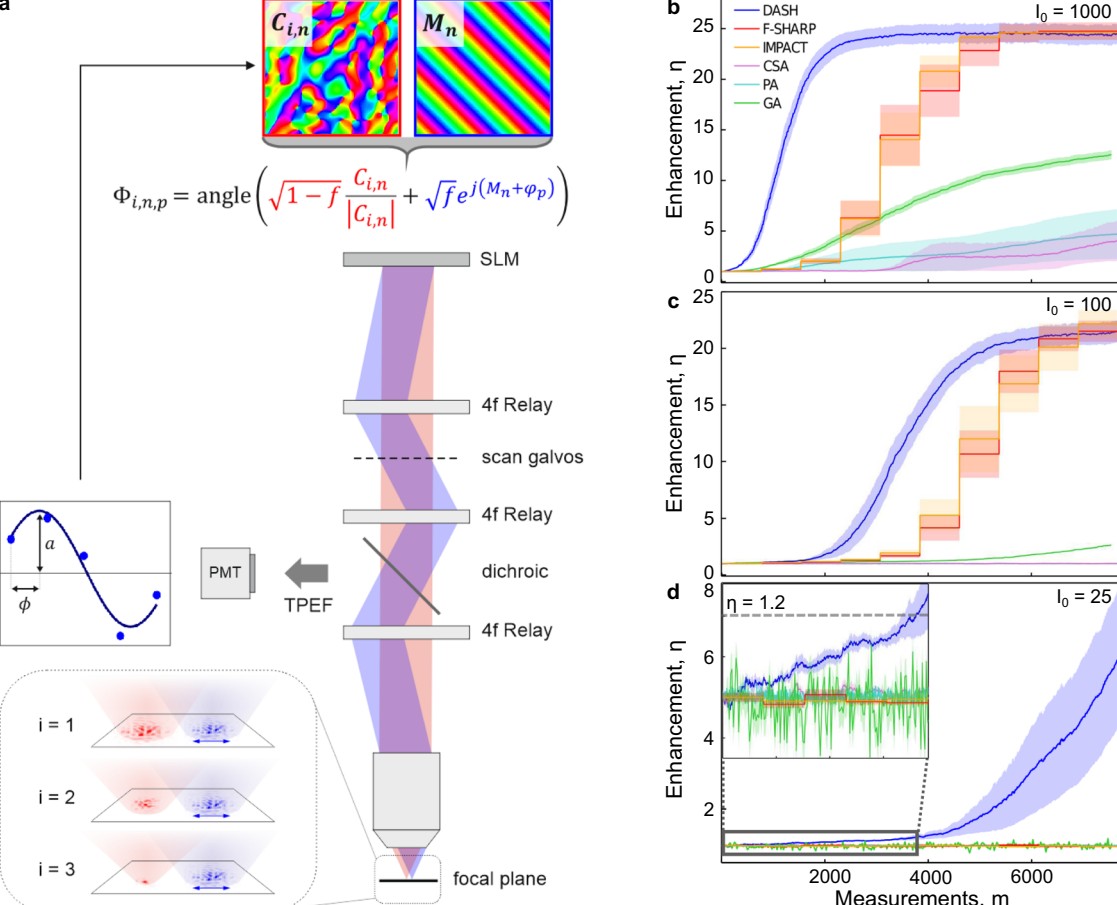

**Fig. 1 Dynamic adaptive scattering compensation holography. a** During DASH, the spatial light modulator (SLM) splits the incident laser beam into a modulated wavefront $M_n$ with a particular phase step $\varphi_p$ and a corrected reference field $C_{i,n}$ with power ratio $f$ as defined by the phase mask $\Phi_{i,n,p}$. Interferometry is then performed by varying the phase step multiple times and recording the corresponding two-photon signals with a photomultiplier tube (PMT). The correction pattern $C_{i,n}$ is then updated by including $M_n$ at its optimal amplitude $a_{i,n}$ and phase $\phi_{i,n}$. As the correction pattern improves, the reference field becomes more point-like after each iteration, i, which improves the accuracy of the phase measurement (inset). **b–d** Simulated comparison of the enhancement $\eta$ achieved by DASH, F-SHARP, IMPACT, a genetic algorithm (GA), a partitioning algorithm (PA), and the continuous sequential algorithm (CSA) for decreasing initial signal levels, $I_0$. The standard error of the mean enhancement over the 10 trials is represented by the color bands around the data and the signal during the first five measurement iterations of part **d** is enlarged in the inset for clarity.

where the two-photon signal will inevitably drop as the imaging depth is increased.

**Experimental demonstration.** The performance of DASH was experimentally compared to F-SHARP by measuring the TPEF signal from a cluster of quantum dots obfuscated by a highly scattering tape layer. The laser was tuned to 810 nm to maximize the two-photon signal and a power of ≤10 mW was used to avoid photobleaching.

TPEF images were acquired at three intervals during the correction process (i) before the first iteration when no correction was available at measurement number $m = 0$, (ii) after the first iteration through the 225 modes corresponding to $m = 1120$, and (iii) when both algorithms have converged after 2025 measurements. During each measurement, the TPEF intensity for 5 subsequent phase steps of $M_n$ is recorded, with a total signal accumulation time per measurement of 1 ms.

The resulting corrected images are shown in Fig. 2a along with intensity profiles extending over 5 μm through the center of the corrected region along the dashed line. A striking difference can be seen after the first measurement iteration, where the DASH correction achieved over an order of magnitude higher signal enhancement than F-SHARP (here, signal enhancement is defined as the ratio of the maximum signal in the corrected image to that in the uncorrected image with the uncorrected scanning beam blocked). At this point, the DASH correction already forms a bright, well-defined focus that could be used for imaging while the F-SHARP correction has hardly improved the wavefront distortion. Furthermore, even after nine measurement iterations when both algorithms had converged, the DASH correction continued to yield a 50% higher signal enhancement than F-SHARP. Note that the color scales of the images in Fig. 2a have been optimized for visibility, while the magnitude of the inset line profiles have been normalized to the maximum DASH corrected signal.

The convergence of the two algorithms was further investigated by measuring the TPEF signal from a uniform layer of dye under a highly scattering tape mask. This uniform sample allows quantification of the signal enhancement from each mode

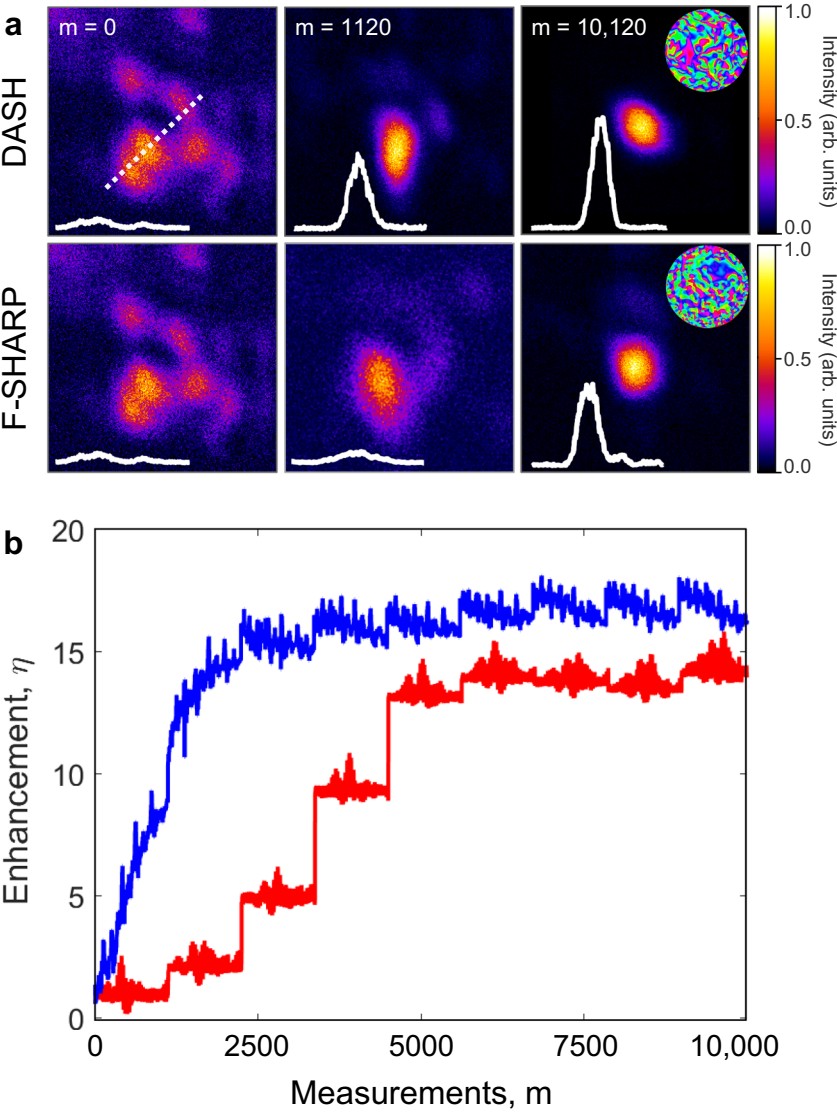

**Fig. 2 Comparison of DASH and F-SHARP. a** TPEF images of a quantum dot sample in DASH (top) and F-SHARP (bottom) before correction, after one iteration, and after both algorithms have converged with intensity profiles extending over a lateral distance of 5 μm along the dashed line and the final phase masks shown as insets along with the measurement number, m. **b** Signal enhancement η on a uniform dye sample after each mode measurement during DASH (blue) and F-SHARP (red) algorithms.

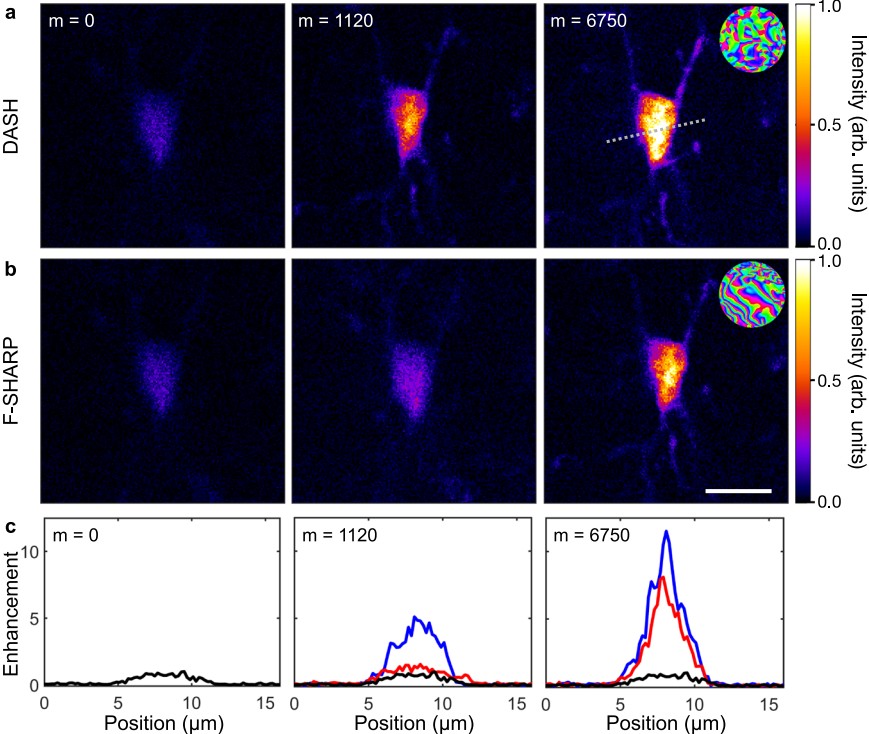

**Fig. 3 Scattering correction in mouse hippocampal tissue. a, b** Images of a single microglia cell in mouse hippocampal tissue acquired before correction ($m = 0$), after the first iteration ($m = 1120$), and after convergence of the DASH and F-SHARP algorithms ($m = 10,120$), respectively, with the resulting correction masks shown in the inset and scale bar corresponding to 10 μm. Similar results were reproduced five times. **c** Profiles along the gray, dashed line in part **a** for no correction (black), DASH correction (blue), and F-SHARP correction (red).

measurement, which is plotted in Fig. 2b. This emphasizes that the DASH algorithm benefits from rapid signal increase with every measurement, enabling a signal enhancement of nearly 11x after just the first iteration. In contrast, the F-SHARP correction is only implemented at the end of each measurement iteration, leading to a step-wise signal increase. This hinders the signal growth significantly, and at the end of the first measurement iteration the TPEF signal is only slightly enhanced. At this stage, the enhancement provided by the DASH algorithm was more than a factor of 8 higher than that provided by F-SHARP. After nine iterations through the correction modes, both algorithms had converged to their optimal correction phase masks and the DASH correction performed only modestly better than the F-SHARP correction.

The higher overall signal enhancement of DASH probably arises from errors introduced due to the experimental complexity of F-SHARP. Specifically, the use of an external scan mirror in F-SHARP means that the correction pattern derived from the scanned interferograms must be manually aligned to the phase mask on the SLM. In contrast, DASH has a common-path design, which means that mode wavefront $M_n$ and reference field $C$ both originate from the same SLM pattern. Each mode $M_n$ is directly applied to the SLM during both the phase stepping interferometry and correction steps, which alleviates the need for an alignment calibration. DASH was also compared experimentally with IMPACT, yielding similar results as discussed in Supplementary Materials Section 2.

**Biological imaging**. To further demonstrate the power of the DASH algorithm, we applied it to correct aberrations and scattering deep inside mouse hippocampal tissue. In these experiments, imaging was performed on resident, resting Cx3cr1$^{GFP}$ microglia in 600 μm thick coronal slices containing the

hippocampus with excitation at 900 nm (see Supplementary Section 6 for details). The accumulation time was increased to 5 ms per measurement, and the laser power on the sample did not exceed 25 mW to avoid tissue damage or photobleaching. Figure 3a shows TPEF images of a single microglia cell at a depth of 350 μm before correction ($m = 0$), after the first iteration ($m = 1120$), and after convergence ($m = 10,120$) for the DASH algorithm (a) and the F-SHARP algorithm (b). While DASH results in somewhat higher signal enhancement than F-SHARP after full convergence, the difference between the two algorithms is most significant after one measurement iteration where DASH provides about five times more enhancement than F-SHARP. The difference in performance is further illustrated in Fig. 3c, which shows the signal profile along the dashed, white line in part (a) for no correction (black), F-SHARP correction (red), and DASH correction (blue) for $m = 0$, 1120, and 10,120.

Finally, we applied the algorithms to image microglia at a depth of 530 μm where the effects of scattering are dominant. In this regime, the F-SHARP algorithm did not converge, while DASH converged after just three iterations ($m = 3370$), yielding a signal enhancement of seven times the maximum value before correction. The resulting TPEF images are shown in Fig. 4 before correction (a) and with the DASH correction (b). Note that the images in Fig. 3 are scaled to give the best contrast, but the amplitudes of the inset profiles can be directly compared.

**Equivalence of IMPACT and F-SHARP**. To provide context for the development of DASH, we present a conceptual framework for understanding the close relationship between IMPACT and F-SHARP, which is validated by numerical simulations. We refer to the Supplementary Section 2 for a rigorous derivation of these observations.

The core idea of IMPACT is that many pixels are simultaneously modulated with unique frequencies, such that their fields produce corresponding signal oscillations when interfering with a static reference wave behind the scattering object. For the correct selection of the modulation frequencies, a single Fourier transform results in a frequency spectrum that contains both the desired phases and low frequency mixing terms as sketched in Fig. 5. Subtracting these phases from the mask on the SLM concludes one step of the algorithm and improves the wavefront quality.

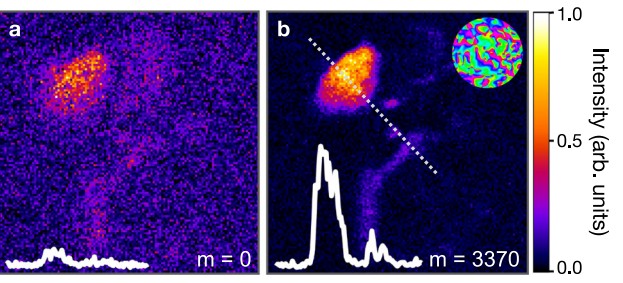

**Fig. 4 Deep tissue imaging.** TPEF images of a single microglia cell at a depth of 530 μm in mouse hippocampal tissue before correction (**a**) and with DASH correction after three iterations ($m = 3370$) (**b**). Intensity profiles extending over 20 μm along the dashed line are shown as insets, along with the final correction mask and similar results were reproduced five times.

On the other hand, the working principle of F-SHARP is to scan a weak external beam across a stronger static field that is corrected by the SLM. Due to the nonlinear signal generation, the stronger SLM field can be assumed to take a spatially more confined shape behind the scatterer and thus takes the role of a "probe beam" measuring the weaker scanning field. Amplitude and phase at each position of the scanning field are retrieved by an interferometric phase stepping procedure and the final correction phase is contained in the phase part of this field, after numerical propagation to the SLM plane.

We note that F-SHARP can be interpreted as a particular implementation of IMPACT. This becomes apparent by examining the equivalent pupil phase distribution of the scanning beam in F-SHARP, which is a tilted plane whose slope changes over time. The set of tilted wavefronts, which create the scanning beam in F-SHARP, are equivalent to unique, equidistant frequency modulations of the pupil field at the discrete positions of the SLM pixels. Fundamentally, the methods are thus equivalent and only differ in some practical aspects, for instance in how the reference wave is generated (externally via a beam-splitter as in F-SHARP or using a portion of static pixels as in IMPACT).

This observation is further validated by the nearly identical convergence of F-SHARP and IMPACT in the numerical simulations shown in Fig. 1.

Like IMPACT, our method relies on a fast SLM, because every new correction pattern is derived from the immediately preceding measurement. Although F-SHARP avoids this problem by employing a high speed external scanning device, we note that

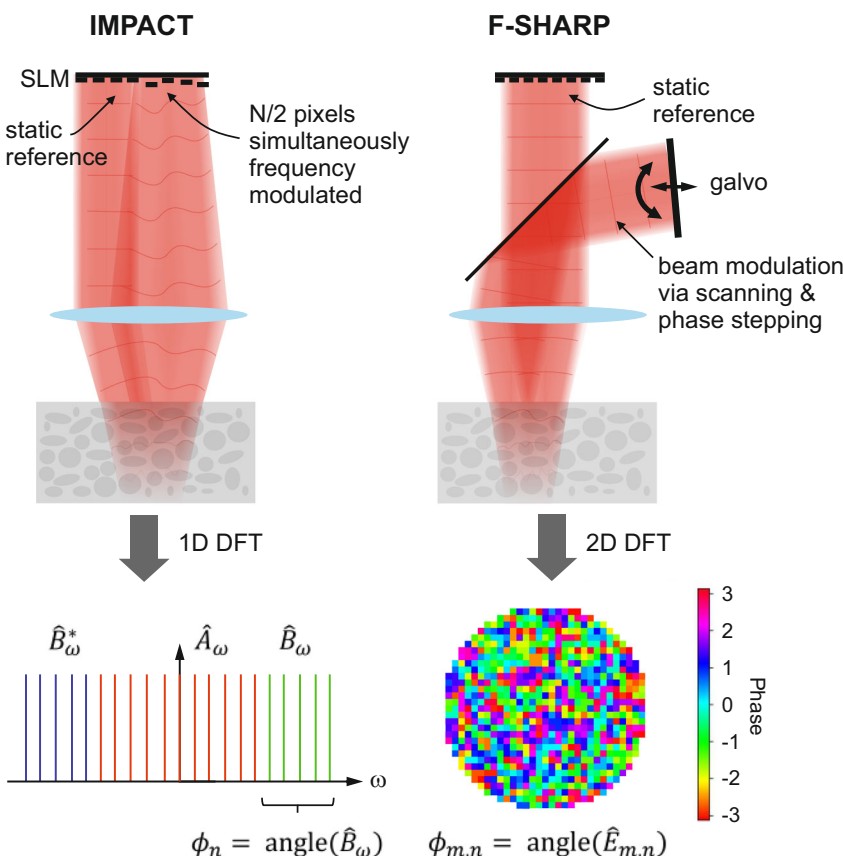

**Fig. 5 Equivalence of IMPACT and F-SHARP. a** Schematic working principles; IMPACT modulates half of the spatial light modulator (SLM) pixels, each with a unique frequency while light from the other, static half acts as reference wave. A 1D discrete Fourier transform (DFT) of the square root of the recorded two-photon signal separates the modulated pixels' correction phases $\phi_n$ of the spectrum $\hat{B}_\omega$ from the contributions of $\hat{B}_\omega^*$ and the mixing terms $\hat{A}_\omega$. F-SHARP relies on external scanning and phase stepping mechanisms to deliver an estimate of the scattered field $\hat{E}_{m,n}$ in the focal region. The correction phases $\phi_{m,n}$ of the first iteration are obtained by a 2D DFT.

very fast SLMs such as MEMs mirrors or acoustic SLMs[36,47] with switching times on the order of microseconds exist. It is also expected that future technological developments will provide even more suitable devices at decreased costs, thereby increasing the importance of immediately updating methods like DASH. The details of the measurement times for our experiments and achievable measurement times for several types of existing SLMs are discussed in Supplementary Section 1.

Furthermore, it is often neglected that the speed of nonlinear biological imaging in deep regions is often limited by low signal levels, requiring dwell times on the order of 100 μs or more. This means that the gains from high speed modulation are limited, while instead it is essential to minimize the number of measurements (and photons) required for an effective scattering correction. In this case, the efficient use of the available photon budget in DASH becomes a significant advantage.

In this work, we introduce a new dynamic algorithm for scattering compensation in nonlinear biological imaging which provides an order of magnitude higher signal enhancement after one measurement iteration compared to existing indirect sensing methods and converges with half as many measured photons. Given the importance of convergence speed and photon budget for imaging through highly scattering, dynamic living tissues, this development is an important step toward real-time deep tissue imaging in a clinical environment and sets the stage for further developments in the field of scattering compensation.

## Methods

A home-built two-photon scanning microscope with tunable femtosecond laser excitation (Spectra Physics Mai Tai DeepSee) is used for TPEF. As shown in Fig. 1a, the excitation beam is sent through a beam splitter (BS, Thorlabs BS014) and the reflected beam is sent to an SLM (Hamamatsu X10468-07), which is imaged onto the entrance pupil of a water immersion objective lens (OL, Olympus XLUMPLFLN20XW, NA = 1) and the xy-scan galvos (Thorlabs GVS011) using 4f relays. Note that while a high resolution SLM display was used here, in general the number of SLM pixels does not need to exceed the number of corrected modes.

The light transmitted through the beam splitter (about a third of the total power) is blocked while running the DASH algorithm, but for the F-SHARP experiments it is reflected off a combined tip/tilt and phase stepping scan mirror (Physik Instrumente S-325) to generate the scanning interferometer arm[38]. The laser power is increased slightly during the DASH algorithm so that the same total power is used for each approach. The excitation light is then either passed back through the beam splitter in the DASH configuration, or the scan and reference arms are interfered on the beam splitter in the case of F-SHARP. Finally, a dichroic mirror (DM) is used to direct the excitation light through the objective to the sample plane and subsequently to send the filtered fluorescence signal to the detection path where it is measured using a PMT (Hamamatsu H10682-210).

The principle of the DASH algorithm is outlined in Fig. 1a and the simulation code is included in the Supplementary Materials. The excitation beam is holographically split by the SLM into a modulated wavefront $M_n$ with an additional phase shift $\varphi_p$ and a reference field $C_{i,n}$, where $i$ denotes the iteration index, starting at $i = 0$, and $n$ is the mode number, which is stepped from 0 to $N-1$. During the first measurement, the phase of the reference field is set to zero, i.e., angle$(C_{0,0}) = 0$. The two fields are given a specific intensity weighting, determined by the constant $f$, by displaying the following phase pattern on the SLM: ssssq

$$\Phi_{i,n,p} = \text{angle}\left(\sqrt{1-f}\,\frac{C_{i,n}}{|C_{i,n}|} + \sqrt{f}e^{j\left(M_n+\varphi_p\right)}\right). \tag{1}$$

The choice of $f$ can influence the convergence behavior, and its optimal value depends on both the sample structure and the signal intensity. Further details on the optimal value of $f$ are provided in Supplementary Section 3, but in practice a value of $f \approx 0.3$ has proven to be robust for all cases that we have investigated in both simulations and experiments.

The wavefront modulations $M_n$ can take the form of a broad range of basis functions. Here, a phase grating basis was chosen with $M_n = k_{x,n}x + k_{y,n}y$, where $x$, $y$ are the row/column pixel indices of the SLM and $k_{x,n}, k_{y,n}$ are the k-vectors of grating $n$. Because the SLM is in a Fourier conjugate plane to the sample (conjugated to the objective pupil plane), the phase gratings effectively scan the modulated beam across the reference beam which is analogous to the scanning interferometry used in F-SHARP. Similarly to F-SHARP and IMPACT, during the initial phase measurements both the reference and modulated beams are scattered by the sample and appear as speckle patterns in the image plane. However, as the correction improves, the reference beam becomes a single bright focus, thereby improving the accuracy of the subsequent phase measurements (see Fig. 1a).

The phase shift $\varphi_p = p(2\pi/P)$ is stepped between subsequent measurements $p = [0, 1, \ldots, P-1]$ of the two-photon signal intensity. The minimum value for $P$ is three. Here, the number of phase steps $P = 5$ was chosen to optimize the correction with the minimum number of measurements under our experimental conditions. A phase stepping interferometry algorithm is then used to determine the phase offset $\phi_{i,n}$ and amplitude weighting $a_{i,n}$ for mode $M_n$ in iteration $i$ as described in the Supplementary Section 4. Finally, this information is immediately used to update $C$:

$$C_{i,n+1} = C_{i,n} + a_{i,n}\,e^{j(M_n-\phi_{i,n})}. \tag{2}$$

This way, the conjugated scattered wavefront is built up step by step. The process is repeated until all of the modes, in our case $N = 225$, have been measured, at which point another iteration through the modes can begin with $M_0$. The final reference field $C_{i,N-1}$ of the completed iteration $i$ acts as initial field for the following one, i.e., $C_{i+1,0} = C_{i,N-1}$.

Apart from the fact that the DASH routine updates the correction mask after each mode measurement, there are two other notable differences to F-SHARP: Firstly, in DASH the mode wavefront $M$ and the reference beam $C$ are shaped by a pure phase mask while in F-SHARP the full complex field is modulated by the beam splitter and scanning mirror. The phase only light modulation used in DASH inevitably introduces a small systematic error to the measured phase. Secondly, each newly measured mode contribution is added to $C$ instead of replacing the contribution found in the previous iteration, which was found to be more robust to errors in the phase measurement in numerical simulations because it effectively averages all previous mode measurements and increases the accuracy of the contribution.

## Data availability

All source data are provided with this paper for Figs. 1–3 at https://github.com/mollyamay/Dynamic-Adaptive-Scattering-Compensation-Holography. Images and Supplementary Data are available from the authors upon request.

## Code availability

Source code written in Julia Version 1.5.3 as well as a simplified example in Python are provided with this paper. The code generates a random phase scatterer and simulates its correction using the different algorithms as shown in Fig. 1a. Images were analyzed in ImageJ Version 1.51.

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

## Acknowledgements
The authors acknowledge funding from the Austrian Science Fund (FWF) (P32146-N36, I3984).

## Author contributions
A.J., M.R.M., and M.K. conceived the experiment. M.A.M. and A.J. built the instrument. M.A.M. performed the measurements. A.J. and N.B. performed the numerical simulations. K.K.K. made the samples.

## Competing interests
The authors report no competing interests.
