## [Peer Review File · Nature Communications]

REVIEWER COMMENTS

Reviewer #1 (Remarks to the Author):

In this manuscript, May et al. describe an interesting iterative feedback based wavefront shaping algorithm, DASH, that can speed up the optimization process and enhance the SNR at each step of the wavefront measurement per incident mode. The method is very interesting and their results also support their technical claims. I believe that the method can potentially be very useful for real applications as can be seen in their experiments through brain slice tissues.

However, I have some questions about their claims about the novelty of the approach. In the manuscript, all comparisons are given with respect to F-SHARP and IMPACT which are based on ideas to measure the entire feedback signal sequence for all modes before moving on to the next iteration (and applying a corrected reference field). F-SHARP and IMPACT are methods that have advantages in other aspects: F-SHARP is fast and can measure large number of modes without fast SLMs. IMPACT requires a fast SLM but the setup is simple and can be easily realized with SLMs with only a small number of pixels (which is usually the case for fast MEMS SLMs). In short, the comparison with DASH is only made with respect to the lack of continuous reference update while their main advantages are in other areas.

The novelty of DASH is not in the wavefront measurement scheme but on the continuous update of the reference field. However, as also noted in the manuscript, there are numerous papers on different wavefront shaping schemes with many of them already continuously updating the reference field [ref 46]. My thought is that a fair comparison should rather be given with continuous sequential algorithms. For example, figures 5&6 in [ref 46] is analogous to the comparisons given in this manuscript with F-SHARP and IMPACT. A comparison with other continuous algorithms instead would greatly help the readers in understanding the advantages of DASH.

In this respect, [ref 41] compares the genetic algorithm with continuous sequential and random partitioning algorithms under different levels of noise. I suggest a similar comparison be made with previous demonstrated continuous algorithms (GS, continuous sequential).

Another important claim of the manuscript is Eqn. 1. Here, the authors claim that complex sum of weighted input modes and reference can be more efficient in speeding up convergence. The method is essentially a way to dynamically increase the reference field relative intensity when the SNR is not sufficient. I like the fact that this can be done with no moving parts and can be changed easily as a part of the algorithm. Perhaps focusing on the effects of the value of f for different SNRs or scattering can be an interesting comparison. Currently, Figure S2 shows only a simple simulation regarding this claim.

Minor points:

1. After convergence, DASH and F-SHARP should show the same signal level. The authors interpret the lower F-SHARP signal level due to numerical misalignment between the galvo and SLM planes. I agree with this interpretation, however, this will have an ill-effect on all F-SHARP results and cause a bias on all figures. It is repeatedly emphasized that after 1 iteration, DASH is 8 times higher than F-SHARP, but this is not a true factor. Why not compare with IMPACT or another method which does not have this issue?
2. a_n in eqn 2 is not described in the main text.

3. In the second paragraph of page 3. $P=5$ phase steps were chosen, “with the minimum number of measurements under our experimental conditions’. What does this mean?
4. In Figure 4, simulations are compared for low SNR scenarios. In fig 4(b), DASH at low SNR shows negligible enhancement for up to 4 iterations, and after 5 iterations, the enhancement shows a threshold behavior. DASH should continuously increase the SNR, why does the efficiency enhancement show this threshold behavior?

Reviewer #2 (Remarks to the Author):

This paper presents a technique to form a tight focus deep within tissue for improved in vivo two-photon imaging. The technique is based upon a repeated update procedure for wavefront shaping that relies on an interferometric measurement of the excited fluorescence as a function of newly added plane wave modes. Experimental results show that the technique can effectively focus over 0.5 mm deep within a coronal slice of mouse hippocampus. Both the theory and experiments are sound and there are sufficient references.

In general, I think this paper is clearly written and introduces a potentially useful technique that is applicable in deep tissue imaging experiments. It is one of a suite of new wavefront shaping tools for multi-photon imaging that can push to deeper imaging depths with sufficient resolution and photon budget. The technique is similar to two other methods: F-SHARP and IMPACT, which the paper carefully reviews. F-SHARP is likely the closest, which appears to measure many modes via galvo scanning, and then apply globally updating. This strategy instead updates mode by mode, which is a subtle but apparently useful addition. I'd therefore argue for publication in Nature Communications, as such an insight can demonstrably improve the quality of wavefront-shaping-based two-photon imaging in deep tissue.

There are some points that I think the authors should address, however, before this work is published:

- 1) The mode-by-mode update strategy presented here is actually very similar to the other “feedback-based” wavefront shaping methods, as first demonstrated by Vellekoop and Mosk in Ref. 18. In this work, it is implemented in a wavevector basis instead of a spatial basis, but the general idea is very similar. I would suggest that the authors explain exactly how their technique differs from these prior (kind of original) approaches a bit more.
- 2) The statement before section 3, “if desired, another iteration through the modes begins with M_0 ” is a bit confusing. I would imagine that just one iteration would not yield very much enhancement. If it did yield effective enhancement, then one of the point-spread functions (scanned or unscanned) would be perfectly sharp, allowing for accurate sampling of the other. This must not be the case with any non-negligible amount of scattering, so I would guess that multiple iterations would nearly always be required. The authors should perhaps clarify if/why the technique still would work well with just 1 iteration through the modes.
- 3) Something else that is confusing is the fact that the mode update occurs in a Fourier conjugate

plane with respect to the image plane. Accordingly, the real picture is that there is an aberrated point-spread function, and the technique makes a separate and slightly weaker aberrated point-spread function that is scanned spatially while modes are established. The ray diagrams in Fig. 1 do not make this clear (it is helpful to have 2 rays per path to clearly show conjugate and Fourier conjugate planes), and even a diagram of the interference and scanning process at the focal plane would be useful

4) While I see details on timing in the supplement section 1, I would have to imagine that this technique in these demonstrations required a rather lengthy amount of time (e.g., with 600-2000 mode updates in Figs 1-3). How long did it take to 1) find the correction pattern and then 2) scan out the image in these experiments? In practice, F-Sharp was quite fast, maybe requiring several seconds to establish a correction pattern, which then allowed the technique to acquire in-vivo images at comparable depths even in the presence of decorrelation from a living/moving organism. A long measurement time might be a downside of the presented technique, and it would be helpful to give the reader a clear picture of this.

Reviewer Responses for Manuscript: Fast holographic scattering compensation for deep tissue biological imaging

We thank all reviewers for their in-depth reports and valuable input. We have implemented extensive changes based on your suggestions that significantly improved the manuscript.

The reviewer comments are shown below in *italics* along with our point-by-point responses in plain text. New and updated text in the manuscript has been highlighted in blue and is also included below for convenience, with deleted text shown in red. Four new figures have also been added to the paper including the updated Fig. 1 shown below and supplementary figures: Fig. S2, Fig. S4, and Fig. S5.

Reviewer 1 (Remarks to the Author:)

In this manuscript, May et al. describe an interesting iterative feedback based wavefront shaping algorithm, DASH, that can speed up the optimization process and enhance the SNR at each step of the wavefront measurement per incident mode. The method is very interesting and their results also support their technical claims. I believe that the method can potentially be very useful for real applications as can be seen in their experiments through brain slice tissues.

However, I have some questions about their claims about the novelty of the approach. In the manuscript, all comparisons are given with respect to F-SHARP and IMPACT which are based on ideas to measure the entire feedback signal sequence for all modes before moving on to the next iteration (and applying a corrected reference field). F-SHARP and IMPACT are methods that have advantages in other aspects: F-SHARP is fast and can measure large number of modes without fast SLMs. IMPACT requires a fast SLM but the setup is simple and can be easily realized with SLMs with only a small number of pixels (which is usually the case for fast MEMS SLMs). In short, the comparison with DASH is only made with respect to the lack of continuous reference update while their main advantages are in other areas.

The novelty of DASH is not in the wavefront measurement scheme but on the continuous update of the reference field. However, as also noted in the manuscript, there are numerous papers on different wavefront shaping schemes with many of them already continuously updating the reference field [ref 46]. My thought is that a fair comparison should rather be given with continuous sequential algorithms. For example, figures 5&6 in [ref 46] is analogous to the comparisons given in this manuscript with F-SHARP and IMPACT. A comparison with other continuous algorithms instead would greatly help the readers in understanding the advantages of DASH.

In this respect, [ref 41] compares the genetic algorithm with continuous sequential and random partitioning algorithms under different levels of noise. I suggest a similar comparison be made with previous demonstrated continuous algorithms (GS, continuous sequential).

This is an excellent point, extensive simulations have been added to the revised manuscript comparing DASH to several established continuously updating algorithms. The comparison was made to a continuous sequential algorithm, a random partitioning algorithm, and a well-established genetic algorithm for three different noise levels. The results highlight the fast and robust convergence of DASH, especially for low signal levels like those in most biological imaging experiments, as shown in the new Fig. 1b-d, and described in detail starting at line 280 of the text.

In reference to the advantages of F-SHARP and IMPACT outside of their updating schemes, a detailed comparison of the measurement speeds of DASH and F-SHARP and an analysis of the situations where F-SHARP has an experimental advantage has been added to Supplementary Section 1 (and copied in response to Reviewer 2, question 4 at the end of this document). Additionally, we would like to clarify that the experimental setup and number of SLM pixels required for DASH is actually identical to that for IMPACT, i.e. DASH does not require more SLM pixels than corrected modes. This point has been added to the manuscript.

New Text Line 280: DASH was benchmarked against five other algorithms using numerical simulations for three different integration times corresponding to decreasing signal levels as shown in Fig. 1(b)-(d). The comparison was made to F-SHARP, IMPACT, and three established continuously updating algorithms including a genetic algorithm (GA) following the procedure in Ref. [41], and the partitioning algorithm (PA) and continuous sequential algorithm (CSA) described in Ref. [46]. Details about the implementation of the GA are provided in Supplementary Section 5.

All simulations assume $N_{\text{scat}} = 1024$ scattering and $N = 256$ correctable pixels. Each method is tested on the same set of 10 different white noise random scatterers located in the Fourier plane. The standard error of the mean enhancement over the 10 trials is represented by the color bands around the data. The sample is a two-dimensional homogeneous fluorescent layer in the focal plane of the objective lens. An ideal detector is assumed, with no readout noise and a quantum efficiency of 100%.

While all of the algorithms achieve some signal enhancement at the highest initial signal level of $I_0 = 1000$ photons/measurement as shown in Fig. 1(b), the signal rises much more quickly during DASH than for the other algorithms and converges to a significantly higher enhancement than the other continuously updating algorithms. F-SHARP and IMPACT converge more slowly, but ultimately to the same enhancement level as DASH. Notably, they

Figure 1 | Dynamic adaptive scattering compensation holography. (a) During DASH, the SLM splits the incident laser beam into a modulated wavefront M_n with a particular phase step φ_p and a corrected reference field $C_{i,n}$. Interferometry is then performed by varying the phase step multiple times and recording the corresponding two-photon signals. The correction pattern $C_{i,n}$ is then updated by including M_n at its optimal amplitude $a_{i,n}$ and phase $\phi_{i,n}$. As the correction pattern improves, the reference field becomes more point-like which improves the accuracy of the phase measurement (inset). (b)-(d) Simulated comparison of DASH to F-SHARP, IMPACT, a genetic algorithm (GA), a partitioning algorithm (PA), and the continuous sequential algorithm (CSA) for decreasing initial signal levels, I_0 . The standard error of the mean enhancement over the 10 trials is represented by the color bands around the data and the signal during the first five measurement iterations of part (d) is enlarged in the inset for clarity.

exhibit significantly higher variance in their enhancements before convergence as evidenced by the by the broad error bands on these data sets, indicating a higher sensitivity to initial conditions like the structure of the scattering mask.

For lower initial signal levels of $I_0 = 100$ photons/measurement as shown in Fig. 1(c), the continuously updating GA, PA, and CSA algorithms do not converge while DASH, F-SHARP, and IMPACT converge after ~ 10 iterations. The poor performance of the continuously updating algorithms for low photon numbers significantly decreases their utility for biological experiments where the two-photon signal will inevitably drop as the imaging depth is increased.

Finally, the algorithms were simulated for even lower signal levels with $I_0 = 25$ photons/measurement as shown in Fig. 1(d), with re-scaled data from the first five measurement iterations shown in the inset. In this case, DASH was the only algorithm to achieve a correction, and a similar enhancement to the high signal cases was reached after about 30 measurement iterations as shown in Supplementary Fig. S5. This result is explained by the fact that DASH efficiently utilizes the information from each measured photon, requiring significantly fewer photons for convergence than IMPACT and F-SHARP as detailed in Supplementary Section 1.

New Text Line 189: Note that while a high resolution SLM display was used here, in general the number of SLM pixels does not need to exceed the number of corrected modes if a pinhole is used to block higher diffraction orders in the sample plane.

Another important claim of the manuscript is Eqn. 1. Here, the authors claim that complex sum of weighted input modes and reference can be more efficient in speeding up convergence. The method is essentially a way to dynamically increase the reference field relative intensity when the SNR is not sufficient. I like the fact that this can be done with no moving parts and can be changed easily as a part of the algorithm. Perhaps focusing on the effects of the value of f for different SNRs or scattering can be an interesting comparison. Currently, Figure S2 shows only a simple simulation regarding this claim.

The ability to tune the power ratio of the modulated and reference waves f is indeed an interesting aspect of the DASH algorithm. In addition to the simulations of how sample structure affects f , we have now investigated the optimization of f on an extended fluorescent sample for increasing initial signal levels as shown in the new Fig. S3. This revealed that while $f = 0.3$ is optimal for low initial signals, its value should be decreased for higher signal levels. These results are carefully discussed in Supplementary Section 3 as shown below.

New Text Supp. Line 126: For an extended fluorescent sample, the optimal value of f depends on a trade-off between minimizing f to favor a focus

determined by the constant reference wave and maintaining enough signal contrast between modulated and reference beams to make an accurate phase measurement. For this reason, the optimal value of f is determined by the initial signal level as shown in Fig. S4(a), in which the enhancement after one simulated DASH iteration was investigated as a function of f for increasing initial signals from a uniform 2D fluorescent layer. For the lowest signal (540 photons/measurement), the enhancement is optimized around $f = 0.3$, while for the highest signal (540k photons/measurement) the enhancement improves down to the smallest simulated ratio $f = 0.01$. This behavior is illustrated in Fig. S4(b)-(c) where the enhancement for each measurement in the low signal and high signal cases is plotted for three increasing values of f .

Minor points:

1. After convergence, DASH and F-SHARP should show the same signal level. The authors interpret the lower F-SHARP signal level due to numerical misalignment between the galvo and SLM planes. I agree with this interpretation, however, this will have an ill-effect on all F-SHARP results and cause a bias on all figures. It is repeatedly emphasized that after 1 iteration, DASH is 8 times higher than F-SHARP, but this is not a true factor. Why not compare with IMPACT or another method which does not have this issue?

This point is well taken. Because the specific factor by which DASH outperforms the other algorithms will vary due to differences in initial conditions like the signal level and structure of the scatterer as well as the precision of the F-SHARP mask alignment, the quantitative comparison of the enhancements can only serve as a rough benchmark for the performance of the algorithms. This point has been clarified in the text, as shown below, and the comparison has been generalized to include IMPACT as suggested. It is worth noting, however, that a careful calibration was performed to align the galvo and SLM plane during the F-SHARP measurements, so in some sense this calibration error does truly limit the performance of F-SHARP.

Furthermore, we have now performed an experimental comparison of DASH and IMPACT as shown in Supplementary Section 2 starting on line 87.

New Text Line 121: Numerical simulations and experimental evaluations of the DASH algorithm in two-photon excited fluorescence microscopy (TPEF) reveal signal enhancement that is roughly 10x higher at the end of the first iteration compared to F-SHARP and IMPACT, although the exact enhancement ratio can vary based on experimental factors like the structure of the scatterer and the accuracy of the F-SHARP interferometer alignment.

Old Text: An experimental evaluation of the DASH algorithm with a continuous Fourier basis in two-photon excited fluorescence microscopy (TPEF) reveals an order of magnitude higher signal enhancement at the end of the first iteration compared to F-SHARP.

New Text Supp. Line 89: The performance of IMPACT was also compared with DASH experimentally and the behavior was similar to that of F-SHARP. In these experiments, IMPACT was implemented as described in Kong, et. al. [3] by modulating a randomly selected half of in total 256 SLM pixels at once. However, as discussed above, modified frequency assignments and phase stepping were used to reduce the required number of measurements.

TPEF images of a 4 μm fluorescent bead under a scattering mask are shown in Fig. S2 with no correction (a), DASH correction (b), and IMPACT correction (c) after ten iterations each, along with the measured correction masks (insets). Similarly to F-SHARP, the final enhancement using IMPACT was somewhat lower than for DASH, as illustrated by the intensity profiles in Fig. S2(d). This could be due to several factors, including the use of power splitting $f = 0.5$ which is implicitly made by modulating half of the pixels, or the choice of a pixel mode basis instead of a Fourier basis.

The signal enhancements during each measurement for the two algorithms are also shown in Fig. S2(e). Here, the IMPACT enhancement is consistently lower than that of DASH because only half of the SLM pixels are corrected during each measurement while the other half are modulated. Note that this does not affect the final correction like that shown in Fig. S2(d), where the optimal phase is applied to all of the SLM pixels.

2. a_n in eqn 2 is not described in the main text.

Thank you for pointing this out, the variable a_n is now described in line 250.

New Text Line 250: A phase stepping interferometry algorithm is then used to determine the phase offset $\phi_{i,n}$ and amplitude weighting $a_{i,n}$ for mode M_n in iteration i as described in the Supplementary Section 4.

3. In the second paragraph of page 3. $P=5$ phase steps were chosen, “with the minimum number of measurements under our experimental conditions”. What does this mean?

While theoretically, $P=3$ phase steps are enough for the interferometric phase measurement, a measurement performed with this minimum number is highly susceptible to environmental factors like time dependent background signals (for example, the 100 Hz flickering of fluorescent lights).

The accuracy of the interferometric measurement can be improved by increasing the number of phase steps, but at the cost of an overall increase in the measurement time. This trade-off was optimized experimentally by performing DASH on a test sample with an increasing number of phase steps and choosing the number ($P=5$) that gave the highest enhancement after 1000 total measurements (where each phase step for each mode represents one measurement).

This choice was further confirmed by fitting the measured two-photon data to a cosine function and increasing the number of phase measurements until the

sum shown in Eqn. 8 of the Supplemental Materials returned the same phase offset as directly fitting the data. A description of this procedure has been added to Supplementary Materials Section 4 as shown below.

New Text Supp. Line 138: It is also worth noting that while theoretically $P=3$ phase steps are enough for the interferometric phase measurement, a measurement performed with this minimum number is highly susceptible to environmental factors like time dependent background signals. The accuracy of the interferometric measurement can be improved by increasing the number of phase steps, but at the cost of an overall increase in the measurement time. Here, this trade-off was optimized experimentally by performing DASH on a test sample with increasing numbers of phase steps and choosing the number ($P=5$) that gave the highest enhancement after 1000 total measurements. This choice was further confirmed by fitting the measured two-photon data to a cosine function and increasing the number of phase measurements until the sum shown in Eqn. 8 returned the same phase offset as directly fitting the data.

4. In Figure 4, simulations are compared for low SNR scenarios. In fig 4(b), DASH at low SNR shows negligible enhancement for up to 4 iterations, and after 5 iterations, the enhancement shows a threshold behavior. DASH should continuously increase the SNR, why does the efficiency enhancement show this threshold behavior?

The DASH enhancement does indeed start to increase even during the first iteration. However, it increases slowly at this point due to the low signal, and the growth in this region is not visible on the scale of the maximum enhancement. We have addressed this by plotting the first five simulated measurement iterations for low signal in the inset of the new Fig. 1d with the appropriate scale.

Reviewer 2 (Remarks to the Author):

This paper presents a technique to form a tight focus deep within tissue for improved in vivo two-photon imaging. The technique is based upon a repeated update procedure for wavefront shaping that relies on an interferometric measurement of the excited fluorescence as a function of newly added plane wave modes. Experimental results show that the technique can effectively focus over 0.5 mm deep within a coronal slice of mouse hippocampus. Both the theory and experiments are sound and there are sufficient references.

In general, I think this paper is clearly written and introduces a potentially useful technique that is applicable in deep tissue imaging experiments. It is one of a suite of new wavefront shaping tools for multi-photon imaging that can push to deeper imaging depths with sufficient resolution and photon budget. The technique is similar to two other methods: F-SHARP and IMPACT, which the

paper carefully reviews. F-SHARP is likely the closest, which appears to measure many modes via galvo scanning, and then apply globally updating. This strategy instead updates mode by mode, which is a subtle but apparently useful addition. I'd therefore argue for publication in Nature Communications, as such an insight can demonstrably improve the quality of wavefront-shaping-based two-photon imaging in deep tissue.

There are some points that I think the authors should address, however, before this work is published:

1) The mode-by-mode update strategy presented here is actually very similar to the other "feedback-based" wavefront shaping methods, as first demonstrated by Vellekoop and Mosk in Ref. 18. In this work, it is implemented in a wavevector basis instead of a spatial basis, but the general idea is very similar. I would suggest that the authors explain exactly how their technique differs from these prior (kind of original) approaches a bit more.

Thank you for this very helpful suggestion, extensive simulations have now been performed to illustrate the relative strengths of DASH in comparison to earlier established feedback based wavefront shaping methods as detailed in the new Fig. 1 and the updated text starting on line 276 and included above in the response to Reviewer 1.

2) The statement before section 3, "if desired, another iteration through the modes begins with M_0 " is a bit confusing. I would imagine that just one iteration would not yield very much enhancement. If it did yield effective enhancement, then one of the point-spread functions (scanned or unscanned) would be perfectly sharp, allowing for accurate sampling of the other. This must not be the case with any non-negligible amount of scattering, so I would guess that multiple iterations would nearly always be required. The authors should perhaps clarify if/why the technique still would work well with just 1 iteration through the modes.

This sentence has been reworded for clarity as detailed below. However, it is worth noting that under many conditions the DASH correction does form a reasonably bright focus that can be used for imaging after just one measurement iteration. For example, in the simulations shown in Fig. 1b, DASH yielded a signal enhancement of nearly a factor of 4 after the first measurement iteration, which is sufficient to suppress the remaining background signal from aberrations and form an image, although the contrast will not be optimal. This is further evidenced by two-photon imaging experiments as shown in Figs. 2 and 3, where a reasonably high imaging quality was achieved after the first measurement iteration ($m = 224$).

In the end, the number of measurement iterations should be decided for a given experiment by balancing the required imaging contrast with the measurement time, which increases linearly with the number of iterations

New Text Line 253: The process is repeated until all of the modes, in our case $N = 225$, have been measured, at which point another iteration through the modes can begin with M_0 .

Old Text: The process is repeated until all of the modes, in our case $N = 225$, have been measured. If desired, another iteration through the modes begins with M_0 .

3) *Something else that is confusing is the fact that the mode update occurs in a Fourier conjugate plane with respect to the image plane. Accordingly, the real picture is that there is an aberrated point-spread function, and the technique makes a separate and slightly weaker aberrated point-spread function that is scanned spatially while modes are established. The ray diagrams in Fig. 1 do not make this clear (it is helpful to have 2 rays per path to clearly show conjugate and Fourier conjugate planes), and even a diagram of the interference and scanning process at the focal plane would be useful.*

Thank you for pointing out this confusion, we have now adapted Fig. 1(a) according to your suggestion. The figure now also shows a schematic of the interference and scanning in the focal plane. We have further added additional explanatory text.

New Text Line 225: The wavefront modulations M_n can take the form of a broad range of basis functions. Here, a phase grating basis was chosen with $M_n = k_{x,n}x + k_{y,n}y$, where x, y are the row/column pixel indices of the SLM and $k_{x,n}, k_{y,n}$ are the k-vectors of grating n . Because the SLM is in a Fourier conjugate plane to the sample (conjugated to the objective pupil plane), the phase gratings effectively scan the modulated beam across the reference beam which is analogous to the scanning interferometry used in F-SHARP. Similarly to F-SHARP and IMPACT, during the initial phase measurements both the reference and modulated beams are scattered by the sample and appear as speckle patterns in the image plane. However, as the correction improves, the reference beam becomes a single bright focus, thereby improving the accuracy of the subsequent phase measurements (see Fig. 1(a))

Old Text: The wavefront modulations M_n can take the form of a broad range of basis functions, but in the comparison to F-SHARP a series of phase gratings was used to scan the modulated beam across the reference beam. In this case, $M_n = k_{x,n}x + k_{y,n}y$, where x, y are the row/column pixel indices of the SLM and $k_{x,n}, k_{y,n}$ the k-vectors of grating n .

4) *While I see details on timing in the supplement section 1, I would have to imagine that this technique in these demonstrations required a rather lengthy amount of time (e.g., with 600-2000 mode updates in Figs 1-3). How long did it take to 1) find the correction pattern and then 2) scan out the image in these experiments? In practice, F-Sharp was quite fast, maybe requiring several seconds*

to establish a correction pattern, which then allowed the technique to acquire in-vivo images at comparable depths even in the presence of decorrelation from a living/moving organism. A long measurement time might be a downside of the presented technique, and it would be helpful to give the reader a clear picture of this.

This is a good point, the innovation in F-SHARP of decoupling the interferometric measurement time from the SLM update rate can be a significant advantage, especially for samples with high two-photon signal and short persistence times. A detailed comparison of the measurement and imaging times under our experimental conditions is now provided in Supplementary Section 1 as shown below.

New Text Supp. Line 35: The initial demonstrations of DASH shown in this work were made with a relatively slow nematic LC-SLM (Hamamatsu X10468-07) with a response time of 50 ms. In this case, nearly 10 minutes were required to optimize the DASH correction mask under the experimental conditions in Fig. 3 of the main text (10 iterations, 224 modes, and pixel dwell time 1 ms). This measurement time is, of course, much longer than the persistence times of most living tissues and therefore only useful for an initial demonstration of the technique. However, the SLM was then upgraded to a faster nematic LC-SLM (Meadowlark HSP1920) with a response time of 3 ms and the measurement time was dramatically reduced to just 45 seconds for the same experimental parameters compared to 22 seconds required for F-SHARP. Notably, DASH has the potential to further improve in speed by more than an order of magnitude if a fast MEMS mirror is used instead of the nematic LC-SLM (see Fig. S1). In this case, the required dwell time would become the rate limiting factor in most biological imaging experiments, and any remaining speed advantage of F-SHARP would be far outweighed by its requirement for higher photon numbers as emphasized in Fig. S1.

REVIEWERS' COMMENTS:

Reviewer #1 (Remarks to the Author):

The authors have addressed all of my questions in the initial submission. I appreciate the new simulations and additional quantitative comparisons and recommend publication.

Reviewer #2 (Remarks to the Author):

I have read through both the authors' response to my comments, as well as the comments from the other reviewer. I think they did a nice job clarifying the various points raised, and I think the added simulations provide a clearer picture of the performance benefits of their technique. The manuscript is now ready for publication in my opinion.

REVIEWERS' COMMENTS

Reviewer #1 (Remarks to the Author):

The authors have addressed all of my questions in the initial submission. I appreciate the new simulations and additional quantitative comparisons and recommend publication.

Response: Thank you for your thoughtful suggestions which improved our paper significantly.

Reviewer #2 (Remarks to the Author):

I have read through both the authors' response to my comments, as well as the comments from the other reviewer. I think they did a nice job clarifying the various points raised, and I think the added simulations provide a clearer picture of the performance benefits of their technique. The manuscript is now ready for publication in my opinion.

Response: Thank you for your thoughtful comments and ideas, which helped us to significantly increase the impact of our work.